# The Effect of Cooling Rates on Thermal, Crystallization, Mechanical and Barrier Properties of Rotational Molding Polyamide 11 as the Liner Material for High-Capacity High-Pressure Vessels

**DOI:** 10.3390/molecules28062425

**Published:** 2023-03-07

**Authors:** Muhuo Yu, Liangliang Qi, Lele Cheng, Wei Min, Zhonghao Mei, Ruize Gao, Zeyu Sun

**Affiliations:** 1State Key Laboratory for Modification of Chemical Fibers and Polymer Materials, College of Materials Science and Engineering, Donghua University, Shanghai 201620, China; 2Center for Civil Aviation Composites, Shanghai Key Laboratory of Lightweight Structural Composites, Donghua University, Shanghai 201620, China

**Keywords:** polymer liner of type IV hydrogen storage vessel, polyamide 11, rotational molding, crystallinity, gas barrier properties

## Abstract

The rapid development of hydrogen fuel cells has been paralleled by increased demand for lightweight type IV hydrogen storage vessels with high hydrogen storage density, which raises the performance requirements of internal plastic liners. An appropriate manufacturing process is important to improve the quality of polymer liners. In this paper, DSC, WAXD, a universal testing machine and a differential pressure gas permeameter were used to investigate the effect of the cooling rate of the rotational molding polyamide 11 on the thermal, crystallization, mechanical and barrier properties. The cooling rate is formulated according to the cooling rate that can be achieved in actual production. The results suggest that two PA11 liner materials initially exhibited two-dimensional (circular) growth under non-isothermal crystallization conditions and shifted to one-dimensional space growth due to spherulite collision and crowding during the secondary crystallization stage. The slower the cooling process, the greater the crystallinity of the specimen. The increase in crystallinity significantly improved the barrier properties of the two PA11 liner materials, and the gas permeability coefficient was 2-3-fold higher than at low crystallinity. Moreover, the tensile strength, the tensile modulus, the flexural strength, and the flexural modulus increased, and the elongation at break decreased as the crystallinity increased.

## 1. Introduction

Polymers are important commercial engineering materials used widely in various applications for their excellent overall properties, such as low density, low cost, high toughness, high modulus, good temperature and corrosion resistance, and excellent fatigue properties [1,2,3,4]. When used for high-pressure vessels and pipelines, polymer liners can provide higher energy storage density and are more cost-effective than metal liners [5,6,7]. Hence, type Ⅳ hydrogen storage vessels (type Ⅳ vessels) with polymer liners have more significant value for onboard hydrogen storage than type Ⅲ vessels [8,9,10,11]. Polymers are typically employed for type IV vessel liners, including PA, HDPE, PET and PU [12,13,14]. Further, PA represents a potential alternative for type IV vessels due to its strong molecular polarity and hydrogen-bond interaction in contrast to the high hydrogen permeability of HDPE [15,16,17].

The thermal, crystallization, mechanical and gas barrier properties of polymer materials determine the hydrogen storage capacity and performance of type IV vessel liners. It has been established that the main factors affecting the service performance of PA are pressure, temperature, time (aging) and material properties [18,19,20]. The pressure and temperature are determined by the external use conditions of polymer materials, while the properties of the materials, such as crystallinity, orientation, free volume, and filler, are determined by processing conditions, the chemical structure and other factors [8,15]. Therefore, many scholars have studied the performance of the liner material. Liu et al. [14] prepared polyethylene/graphene flake (PE/GF) composite films with ultrahigh gas barrier properties by using the unbounded confinement hot pressing method for polymer liners. The results showed that the crystallinity of the composite films increased by approximately 10% after hot pressing, and the free volume fraction decreased from 6.52 vol% to 5.43 vol%. Due to the synergistic effect of the above aspects, the barrier properties of the composite films were significantly improved. In addition, the tensile strength and Young’s modulus of the composite films were increased by 31.4% and 21.1%, respectively. Sun et al. [15] comprehensively studied the suitability of PA6 filled with lamellar inorganic components (LICs) as type IV vessel liners, including thermal and mechanical properties, morphology and structure, rheology, and hydrogen permeability under various temperature and pressure conditions. The results showed that the thermal and processing properties of LIC/PA6 were improved, and the tensile strength, bending strength and the bending modulus of LIC/PA6 increased by 36%, 17% and 12%, respectively, compared with the pure component PA6. Importantly, the hydrogen permeability of LIC/PA6 was decreased by 3–5 fold.

PA11 is a kind of high-performance engineering thermoplastic, which has the characteristics of low water absorption, corrosion resistance, low temperature resistance, good sealing and easy processing [21]. Further, large-capacity hydrogen storage cylinders need to be manufactured by rotational molding, and it is difficult for short-chain PA to meet its stringent processing performance requirements [22]. PA11 are semi-crystalline polymers, of which the physical, chemical and mechanical properties are all strongly affected by crystallinity [23]. The crystallinity of polymers depends not only on molecular structure but also on processing conditions. Among them, changing the processing conditions is a simple and efficient method. The conditions that affect rotational molding mainly include mold temperature, rotation rate and cooling rate [24]. However, the influence of cooling rate on the thermal, crystallization, mechanical, and barrier properties of rotational nylon samples has rarely reported.

In this work, the rotational molding process was used to prepare the polymer liner of type IV hydrogen vessels, and the crystallization properties of the polymer were controlled by changing the cooling method, which was a simple, efficient and economical method. The cooling rate used in this experiment mainly refers to the cooling conditions of the rotational molding process. Rotational molds can be cooled by air or water [24]. Cooling by air is mainly natural cooling and using external conditions such as fans. Water cooling is mainly to spray water on the mold for cooling. Finally, DSC, WAXD, a universal testing machine and a differential pressure gas permeameter were used to investigate the effect of the cooling rate of the rotational molding polyamide 11 on the thermal, crystallization, mechanical and barrier properties.

## 2. Results and Discussion

### 2.1. Thermal Analysis

The DSC curves of the two PA11 materials synthesized at the same heating rate and the same cooling rate are shown in Figure 1. The melting point of PA11 (a) was 169 °C and the melting peak temperature was 177 °C; the crystallization temperature was 152 °C and the crystallization peak temperature was 148 °C. The melting point of PA11(b) was 172 °C with a melting peak temperature of 182 °C, a crystallization temperature of 159 °C, and a crystallization peak temperature of 156 °C.

Since the non-isothermal crystallization process is reportedly closer to the actual process of polymer processing, exploring non-isothermal crystallization kinetics is instrumental in understanding the effect of the temperature field on the polymer crystallization process and crystal structure morphology [25]. Hence, the investigation of non-isothermal crystallization is essential for practical applications. The DSC cooling curves at varying cooling rates (5–15 °C/min), as illustrated in Figure 2, were used to explore the non-isothermal crystallization process. The crystallization onset ($T_{o}$), the peak ($T_{P}$) and the end ($T_{e}$) temperature results are recapitulated in Table 1.

Accordingly, the $T_{o}$ and the $T_{P}$ of all samples shifted to a lower temperature by increasing the cooling rate, which illustrated that more degrees of supercooling were required to activate crystallization and nucleation when the samples were rapidly cooled [26]. Moreover, the peak shape widened as the cooling rate increased [27]. As the cooling rate increased, the crystallization time was shortened, which resulted in insufficient time for the PA11 specimens to crystallize. Thus, smaller crystals with more defects were obtained.

Under non-isothermal crystallization conditions, the overall crystallization time ($t_{c}$) was calculated from the following equation:(1)$$t_{c}=\left(T_{0}-T_{e}\right)/\varphi$$
where φ is the cooling rate. As shown in Table 1, the overall crystallization time was reduced when the cooling rate was more significant. The fluidity of the polymer chains decreased faster, and the formation of crystals was restricted as the cooling rate increased. Therefore, the required crystallization temperature gradient increased, and the corresponding peaks widened [26,28].

Given that the non-isothermal crystallization process is complicated, different theories have been proposed to explain the non-isothermal crystallization kinetics by Ozawa, Jezionrny and Mo [29,30,31,32]. Figure 3 illustrates the relative degree of crystallinity $\;X_{c}\left(T\right)\;$ versus the temperature, which revealed that the relative crystallinity as a function of temperature and all curves exhibited similar sigmoidal shapes. The values of relative crystallinity at different cooling rates $X_{c}\left(T\right)$ were calculated using Equation (2).
(2)$$X_{c}\left(T\right)=\frac{\int_{T_{o}}^{T}\left(dH_{c}/dT\right)dT}{\int_{T_{o}}^{T_{e}}\left(dH_{c}/dT\right)dT}$$
where $dH_{c}/dT$ represents the crystallization heat flow rate, and the $T_{o}$ and the $T_{e}$ are the onset and end of the crystallization temperature.

The relationship between crystallization temperature T and time t can be calculated by:(3)$$t=\frac{\left|T_{o}\;-\;T\right|}{\varphi }$$
where t is the crystallization time, and the $T_{o}\;$ and the T are the temperature at the beginning (*t* = 0) and crystallization temperature, respectively. φ is the cooling rate. The relationship between relative crystallinity $X_{c}\left(t\right)$ and crystallization time t can be transformed into Equation (4).
(4)$$X_{c}\left(t\right)=\frac{\int_{t_{o}}^{t}\left(dH_{c}/dt\right)dt}{\int_{t_{o}}^{t_{e}}\left(dH_{c}/dt\right)dt}$$
where the $t_{o}$ and the $t_{e}$ are the onset and total crystallization times, respectively. The plots of $\;X_{c}\left(t\right)$ versus t for two PA11 specimens at different cooling rates are presented in Figure 4. The tails of all curves flattened, indicating that the crystallization rate slowed since the spherulites collided and squeezed at the end of the non-isothermal crystallization process. This behavior was attributed to the existence of two crystallization processes: a fast “primary” process at the initial stage and a slower “secondary” process at the later stage of crystallization.

Considering the effect cooling rate, the following polymer kinetic equation was derived by Ozawa at a constant cooling rate φ [33].
(5)$$1-X_{c}\left(t\right)=exp\left[-K\left(T\right)/\varphi^{m}\right]\;log[-In(1-X_{c}\left(t\right))]=logK\left(T\right)-mlog\varphi$$
where K(T) is a function related to the nucleation method, nucleation rate, and crystal nucleus growth rate, and m is the Ozawa exponent depending on the dimension of crystal growth. The plot of $log[-In(1-X_{c}\left(t\right))]$ versus logφ at a given temperature was generated. It is generally thought that if Ozawa theory is valid, a straight line should be obtained. As shown in Figure 5, no straight lines were obtained, and the kinetic parameters m and K(T) could not be derived from the slope and the intercept, respectively. The results showed that the Ozawa exponent was not consistent with the temperature during the non-isothermal crystallization, probably because of the inaccurate assumption of Ozawa’s theory, since secondary crystallization was not taken into consideration and the dependence of the fold length on temperature [31,34].

Avrami theory has been widely used to analyze the isothermal crystallization process and describe the primary stage of non-isothermal crystallization [35,36].
(6)$$1-X_{c}\left(t\right)=exp(-Z_{t}t^{n})\;log[-In(1-X_{c}\left(t\right))]=nlog\left(t\right)+log\left(Z_{t}\right)$$
where $Z_{t}$ is a growth rate constant that is dependent on nucleation and growth rate parameters, and n is a mechanism constant value that is dependent on the type of nucleation and the growth dimension.

The plot of $log[-In(1-X_{c}\left(t\right))]$ versus logt was generated using Equation (6), as plotted in Figure 6. The values of the Avrami equation $Z_{t}$ and n extracted from the intercept and the slope of the linear portion of the curves are listed in Table 2. All curves could be divided into initial and secondary crystallization stages, which indicated the existence of secondary crystallization in the non-isothermal crystallization process at 50–60% relative crystallinity for PA11 specimens. The fitting lines of all samples were almost parallel, indicating similar rate constants and Avrami exponents at various cooling rates. As shown in Table 2, during the primary crystallization stage, the Avrami exponents $n_{1}$ was approximately 1.81, 1.92 and 1.97 for PA11 (a), 1.80, 2.02 and 2.03 for PA11 (b), $n_{2}$ was approximately 0.68, 0.83 and 1.06 for PA11 (a), and 0.70, 0.97 and 0.99 for PA11 (b). The n values of both PA11 specimens at various cooling rates ranged from 1.80 to 2.10, indicating that the crystallization model of PA11 specimens under non-isothermal crystallization conditions might involve two-dimensional (circular) growth. During the secondary crystallization stage, the Avrami exponent was close to 1, and the growth form of spherulites was transformed into one-dimensional space extension because of spherulites impingement and crowding. Moreover, the $Z_{t}$ value for all samples increased by increasing the cooling rate, which may be because a higher cooling rate causes melt crystallization at a lower temperature and has a higher crystallization rate because of increased undercooling.

Mo and Liu et al. [31,32,37] elaborated a combined Avrami–Ozawa equation to characterize the non-isothermal crystallization kinetics based on the assumption that the degree of crystallinity is correlated with the cooling rate and crystallization time as follows [38]:
(7)$$\log\left(Z_{t}\right)+nlog\left(t\right)=\log\left(K\left(T\right)\right)\;-\;mlog\left(\varphi \right)\;\log\left(\varphi \right)=\log\left(F\left(T\right)\right)\;-\;blog\left(t\right)\;F\left(T\right)={\left(K\left(T\right)/Z_{t}\right)}^{1/m}\;b=n/m$$
where n is the apparent Avrami exponent during the non-isothermal crystallization process, and m is the Ozawa exponent. F(T) refers to the necessary cooling rate required to achieve a given relative crystallinity at unit crystallization time, which can be used as a parameter for the crystallization rate of polymers. The larger the value of F(T) is, the slower the crystallization rate, and vice versa. According to Equation (7), the plots of log(φ) versus log(t) at a given degree of crystallinity with an intercept of log(F(T)) and a slope of −b are presented in Figure 7 and Table 3, from which an excellent linear correlation can be established. The higher the given percentage of relative crystallinity is, the higher the value of F(T), indicating that the necessary cooling rate should be increased per unit of crystallization time [39]. In addition, the crystallization rate decreased when the relative crystallinity increased under non-isothermal crystallization conditions.

To estimate the effective energy barrier of the non-isothermal crystallization process, the crystallization activation energy (ΔE) for polymer chain migration to the growing surface is usually calculated by the Kissinger equation as follows [28,40].
(8)$$d\left[In\left(\varphi /\left(T_{p}^{2}\right)\right)\right]/d\left(1/T_{p}\right)=-\Delta E/R$$
where R is the gas constant and $T_{p}$ is the crystallization peak temperature. According to Equation (8), the plots of
$In(\varphi /(T_{p}^{2}))$ versus $1/T_{p}$ and ΔE can be calculated from the slopes, ΔE=R×slope. It is well established that polymer chains or segments have a higher viscosity and a lower diffusivity and are high during melting, which must overcome energy barriers to transport and deposit onto the surface of the crystal. Therefore, the crystallization activation energy ΔE is closely associated with the crystallization process, and a lower activation energy suggests that triggering crystallization was relatively easy. The ΔE value for PA11(a) was 242.6 KJ/mol and that of PA11(b) was 299.3 KJ/mol (Figure 8), suggesting that it was more difficult for PA11(b) to crystallize, consistent with our findings on the F(T) value using the Mo method, as described above.

### 2.2. Crystal Structure Analysis

The crystal structure of polyamide consisted predominantly of hydrogen bonds. The hydrogen bond sheet was the main feature of the polyamide structure, which accounted for the two strong reflections of diffractograms. The comprehensive intensity distributions of the WAXD modes of the two PA11 samples at different cooling rates are illustrated in Figure 9. PA11 mainly comprised five crystal forms, including the α phase produced by annealing of quenched polymer or m-cresol solution casting, α’ phase obtained by melt crystallization, pseudo hexahedral phase obtained by heating α’ above room temperature, and obtained by melt quenching smectic phase or δ’ phase and γ phase obtained by casting from trifluoroacetic acid solution [41,42]. All crystallized PA11 samples showed characteristic reflections at 2θ = 20.1° and 23.4°. The amide group was located on a plane inclined to the chain axis according to the α phase with a triclinic unit cell proposed by Slichter [43] and Kawaguchi [44]. The structure was an H-bonded sheet-like structure similar to PA6,6 and hydrogen bonds were formed between parallel chains. The two strong reflection positions were indexed as 100 and 010/110. The d-spacings of these two strong reflections were approximately 0.44 and 0.37 nm, representing the interlamellar distance between the lamellae and the projected interchain distance within the lamellae, respectively. The 001 reflection occurs at 2θ = 7.5°, corresponding to a d-spacing of 1.71 nm, and the 023 reflection at 2θ = 37.9°, corresponding to a d-spacing of 0.23 nm. The crystallinity of the PA11 specimens was calculated based on Equation (1), the crystalline peak correction factors are $C_{001}\left(\theta \right)=0.085$, $C_{100}\left(\theta \right)=1$, $C_{010,110}\left(\theta \right)=1.57$ and $C_{023}\left(\theta \right)=10.5$. The amorphous peak was corrected by a factor of $C_{a}\left(\theta \right)=1.12$ and a total correction factor of $K=K_{i}C_{a}\left(\theta \right)=0.8111$. According to Jade software and WAXD results, the crystallinity of PA11 (a) was 35.30% at 05 °C/min, 28.41% at 10 °C/min and 24.18% at 15 °C/min. The crystallinity of PA11 (b) is 42.81% at 05 °C/min, 39.26% at 10 °C/min, and 35.09% at 15 °C/min. The slower the cooling process, the greater the crystallinity of the specimen since the slow crystallization could make the polymer chains stay within the crystallization state for a longer time to ensure that the polymer chains could be fully discharged into the crystal lattice, thereby increasing the crystallinity. Excessively fast cooling and crystallization caused the polymer chains to froze before crystallization.

To further analyze the effect of different cooling rates on the crystal structure of the PA11 sample, the thickness of the crystal $L_{hkl}$ (coherence length) perpendicular to the reflection surface  hkl, was calculated using the Scherrer equation.
(9)$$L_{hkl}=\frac{K\lambda }{\beta_{0}cos\theta }\;\beta_{0}^{2}=\beta_{M}^{2}-\beta_{I}^{2}$$
where λ is the wavelength of the X-ray (nm), $\beta_{0}$ is the width of the diffracted beam (rad), $\beta_{M}$ is the measured width (rad) of the diffracted beam, $\beta_{I}$ is the instrument width (rad), and K is the Scherrer factor related to $\beta_{0}$ and $L_{hkl}$. When $\beta_{0}$ is defined as the full width at half maximum of the diffraction peak, K=0.9. The calculated relevant $L_{hkl}$ values are listed in Table 4.

As seen in Table 4, the crystal thickness of the $L_{100}$ plane was larger than the $L_{hkl}$ value of other planes, since the hydrogen bonds in the (100) plane molecular chain could easily be moved to lead to recrystallization on the (100) plane, and the activation energy of the recrystallization process was lower. In addition, a correlation between $L_{hkl}$ and the cooling rate was observed, the mobility of polymer chains was poor, and the diffusivity of the chains in the growing crystal was hindered with an increased cooling rate. The average size of the crystallites formed was gradually reduced, and the crystal thickness was smaller.

### 2.3. Mechanical Properties

The experimental curves of the tensile and bending properties of the two PA11 materials are shown in Figure 10. Due to the uneven deformation caused during the measurement of the specimen, the strain rate given to the specimen during the experiment is not constant. The global strain rate in the loading direction was calculated by averaging the local strain rates of the extensometer. The tensile and flexural modulus were obtained by linear regression using nominal stress–nominal strain curves over a strain range of 0.5% to 2.5%. The tensile curve of PA11 (a) revealed a linear relationship during the initial deformation stage, and then the slope gradient and stiffness gradually decreased until ductile fracture occurred. From the tensile curve of PA11 (b), the stress–strain showed a linear relationship in the initial deformation stage. After the maximum stress, the stress decreased with necking until ductile failure. The mechanical test results of two PA11 specimens were shown in Figure 11. The PA11 (a) sample exhibited a tensile strength of 37.67 MPa, a tensile modulus of 0.63 GPa, and an elongation at break of 32.31%; a flexural strength of 55.23 MPa and a flexural modulus of 1.08 GPa at a cooling rate of 05 °C/min. When the cooling rate reached 15 °C/min, the tensile strength decreased by 15.24% to 31.93 MPa, the tensile modulus decreased by 14.29% to 0.54 GPa, and the elongation at break increased by 46.67% to 47.24%; the flexural strength decreased 18.09% to 45.42 MPa, and the flexural modulus decreased 6.49% to 1.01 GPa. Among them, the tensile strength of the PA 11 (b) sample at a cooling rate of 05 °C/min was 44.36 MPa, the tensile modulus was 1.12 GPa, and the elongation at break was 38.90%; the flexural strength was 66.03 MPa, and the flexural modulus was 1.18 GPa. When the cooling rate increased to 15 °C/min, the tensile strength decreased by 5.70% to 41.83 MPa, the tensile modulus decreased by 5.36% to 1.06 GPa, and the elongation at break increased by 8.25% to 42.11%; the flexural strength decreased by 11.65% to 58.34 MPa, and the flexural modulus decreased by 3.39% to 1.14 GPa. As the cooling rate slowed, the crystallinity of the two PA11 samples increased, the molecular chains were arranged in a tight and orderly manner, the porosity was relatively low, and the intermolecular interaction force was enhanced. Accordingly, the strength and the moduli of the materials were improved. At the same time, the crystallinity increased, and the molecular segments had no room for movement, which caused the elongation at break to decrease. In addition, a slow cooling rate could easily produce large spherulites, which had relatively more crystal plane defects, and these defects increased susceptibility to damage after stress and reduced the elongation at break.

### 2.4. Barrier Properties

Figure 12 depicts the gas permeability coefficients of two PA11 materials with different cooling rates. Obviously the order of permeability coefficients for PA11 materials is P (05 °C/min) < P (10 °C/min) < P (15 °C/min). The results showed that under the cooling condition of 15 °C/min, the permeability coefficients of the two materials both reach maximum values, which are 7.23 × 10^−13^ cm^3^·cm/cm^2^·s·Pa and 4.39 × 10^−13^ cm^3^·cm/cm^2^·s·Pa, respectively. Slowing the cooling experimental condition to 05 °C/min, the permeability coefficients of the two materials both reach the lowest values of 3.61 × 10^−13^ cm^3^·cm/cm^2^·s·Pa and 1.74 × 10^−13^ cm^3^·cm/cm^2^·s·Pa, respectively. The barrier properties were increased by approximately 2–3 fold.

As shown in Figure 13a, the permeation behavior of small gas molecules could be described as the dissolution and permeation process of small gas molecules at the gas-solid interface and the diffusion permeation process of gas molecules inside the polymer (solid) material [45]. Among them, the permeation process of small gas molecules inside the polymer involved diffusion from areas of high concentration to low concentration [46]. According to Fick’s law, the permeability coefficient is the product of the diffusion coefficient and the solubility coefficient [47,48]. Factors affecting the diffusion coefficient include free volume and gas permeation paths [49]. When the crystallinity is high, the molecular chains are tightly arranged, which leads to a decrease in the free volume fraction [50]. As for the gas permeation path, gas permeation in a polymer matrix can be described in terms of tortuosity [51], that is, the process by which gas penetrates the surface of the polymer material and passes around internal obstacles such as fillers, as shown in Figure 13b. If tortuosity is applied to a crystalline polymer, the crystalline region in the polymer is the blocking part because the permeation of gas inside the polymer exhibits thermodynamic behavior and must follow the least energy path. After the polymer crystallized, the molecular chains were neatly stacked, and the stacking density was high, forming a permeation path with higher energy, while the loose structure of the amorphous region could easily form a permeation path with lower energy [14]. Indeed, it is challenging for small gas molecules to penetrate the permeable crystal, and the area where the penetration occurs is the amorphous region. Therefore, the high crystallinity of the polymer blocks gas to a certain extent and reduces the internal permeability of the polymer. Therefore, the aggregation structure can be improved by changing the process conditions, and the barrier performance can be improved.

## 3. Experimental

### 3.1. Materials

The material of the polymer liner was PA11(a), XXX with 1.05 g/cm^3^ and PA11(b), XXX with 1.03 g/cm^3^ was used as rotational molding powders. Owing to the high hydrophilicity of polyamide, two samples were kept in a vacuum oven at 105°C for 12 h before use.

### 3.2. Control of PA11 Liner Processing

As shown in Figure 14, rotational molding involves the processing of polymer materials to create large-sized hollow parts. The principle of rotational molding is simple, but the ability to manufacture complex parts is a key factor in its success. During the polymer liner molding process, the metal insert and the liner part can be assembled, and the parts can be manufactured without the defects caused by welding. Accordingly, rotational molding has more advantages in polymer liner molding and manufacturing.

The thermoplastic liner preparation was carried out using a tower rotary rotational molding machine (3A-3500, Rising Sun Rotomolding Machine, Wenling, China). The PA11 liner of rotational molding is shown in Figure 15. Before processing, the internal surface of the vessel mold was covered with a silicone-based release agent to facilitate demolding. The polymer powder was first loaded into the mold and then closed. The amount of polymer powder was calculated in advance to obtain the desired thickness and shape of the sample. The thickness of the designed thermoplastic liner was approximately 4 mm. Accordingly, the required weight of PA11(a) was 10.9 kg, and that of PA11(b) was 10.6 kg. After closing, the mold began to run on two orthogonal axes with a shaft speed ratio of 1:3. The rotating mold was then placed in an oven, where it was heated to the melting temperature of the polymer powder, which then began to melt and gradually formed the inner shape of the mold. Rotation was conducted until the material was evenly distributed to ensure that the thickness of the workpiece was a constant value. Heat was provided by gas or oil burners. The mold temperature was measured by placing a thermocouple in the container mold to control the processing temperature more precisely. The oven temperature was set to 320 °C, the inner surface temperature of the container mold was approximately 230 °C, the heating rate was approximately 10 °C/min, and the holding time was approximately 300 s. Biaxial rotation continued, and the mold was transferred to the cooling stage. The cooling stage was carried out by natural cooling, air cooling and water cooling. When the internal temperature of the mold cooled to approximately 100 °C, the mold was opened to release the mold, and the final sample was removed from the mold. The rotational molding parameters used in this investigation for making the vessel liner are shown in Table 5.

### 3.3. Characterization and Measurements

The thermal properties of PA11 were measured by a differential scanning calorimeter (DSC-214, NETZSCH, Selb, Germany) in the temperature range of 20~230 °C. Due to the influence of sample weight on the experimental results, 5~10 mg of both materials was tested. PA11 powder was heated to 230 °C at a heating rate of 10 °C/min and maintained at this temperature for 5 min to eliminate the previous thermal history. The non-isothermal crystallization behavior of PA11 powder was then investigated by cooling the powder from 230 °C to 20 °C at constant cooling rates of 05, 10 and 15 °C/min.

PA11 films with different cooling rates (05, 10 and 15 °C/min) were prepared by a hot-stage microscope (BX51-P, OLYMPUS, Tokyo, Japan), and the films were cut into squares of approximately 10 mm×10 mm. Wide-angle X-ray diffraction (WAXD, Bruker-AXS D8 ADVANKE, Bruker, Billerica, USA) was used to analyze the crystal structure of PA11 films with different cooling rates. The diffractometer was operated at a generator voltage of 40 kV and an intensity of 150 mA and Cu-K radiation (λ = 0.15406 nm) was used to scan the sample from 5° to 45° (2θ) at a scan speed of 3°/min during the test. All data analyses were performed using Jade software. The crystallinity determined by XRD was calculated using Equation (1):(10)$$W_{cx}=\frac{\sum_{i}C_{i,hkl}\left(\theta \right)I_{i,hkl}\left(\theta \right)}{\sum_{i}C_{i,hkl}\left(\theta \right)I_{i,hkl}\left(\theta \right)+k_{i}C_{a}\left(\theta \right)I_{a}\left(\theta \right)}$$
where $I_{i,hkl}\left(\theta \right)$ is the relative intensity of the crystalline peak, $I_{a}\left(\theta \right)$ is the relative intensity of the amorphous peak, $C_{i,hkl}\left(\theta \right)$ is the correction factor for the crystalline peak, $C_{a}\left(\theta \right)$ is the correction factor for amorphous peaks, and $k_{i}$ is the relative scattering coefficient.

The tensile properties of the specimens were measured using a universal testing machine (3382, Instron, Norwood, USA) and a 100 kN load cell in accordance with ISO 527. A dumbbell-shaped specimen of 165 × 19 × 4 mm^3^ was placed in the fixture of the universal testing machine, and the experimental speed was set to 10 mm/min. The data of 8 samples were tested to obtain the statistical standard deviation, and the tensile strength, Young’s modulus and elongation at break of the samples were obtained from the average value of 8 tensile samples taken from the axial ends of type IV vessels (Figure 3). The bending properties of the specimens were tested by three-point bending according to ISO 178 at a speed of 5 mm/min. The size of the bending test specimen was 4 mm (thickness) × 10 mm (width) × 80 mm (length) and was cut from the middle axial part of both ends of type Ⅳ vessels, and the load span was set to 64 mm. The data for 8 specimens were analyzed to obtain the statistical standard deviation and the mean values used to obtain flexural strength and the flexural modulus (Figure 16).

The permeability test of PA11 was conducted using the differential pressure method according to ISO 2556-1974. The gas leak rate was measured using a differential pressure gas permeameter (VAC-V1, Labthink Instruments, Jinan, China). The polymer film divides the test vessel into two compartments, compartment 1 and compartment 2. The permeate gas was introduced into compartment 1, and gas molecules penetrated the polymer membrane. The polymer membrane was saturated with permeating gas molecules by diffusion. Finally, the permeate gas was transferred from compartment 1 to compartment 2 through the polymer membrane. The ability of a gas to pass through a polymer is described by its permeability coefficient, which is the product of its solubility and diffusion coefficient. Specifically, the gas permeability coefficient (P) of gas passing through the PA11 film was recorded by the time delay method at 25 °C under an upstream pressure of 1 atm using the pressure swing permeation method, and the calculation formula is shown in Equation (2). The sample size was approximately 1 mm thick, and the length and width were approximately 15 mm. The average value of the barrier properties of at least five samples was measured as follows.

(11)$$P=\frac{\Delta p}{\Delta t}\times \frac{V}{S}\times \frac{T_{0}}{T\times P_{0}}\times \frac{D}{\left(P_{1}-P_{2}\right)}$$
where Δp/Δt (Pa/h) is the pressure change rate at steady state, V (cm^3^) is the downstream chamber volume, S (cm^2^) is the effective area of the test sample, T (K) is the temperature at the time of measurement, $p_{0}$ (1.0133 × 10^5^ Pa) and $T_{0}$ (273.15 K) are the standard pressure and temperature, respectively. D (cm) is the thickness of the membrane, and $p_{1}$ and $p_{2}$ represent the pressure between the two sides of the membrane.

## 4. Conclusions

The present study explored the physical and chemical properties of two types of rotational molding polyamide 11 at different cooling rates, focusing on the characterization of the thermal, crystallization, mechanical and barrier properties. The method developed by Mo and Liu combined with the theory of Avrami and Ozawa, successfully described the non-isothermal crystallization process of PA11. The results showed that the two PA11 liner materials might exhibit two-dimensional (circular) growth under non-isothermal crystallization conditions and then transform into one-dimensional space growth due to spherulite collision and crowding in the secondary crystallization stage. The crystallization onset ($T_{o}$) and peak ($T_{P}$) shifted to a lower temperature by increasing the cooling rate for both specimens, illustrating that more degrees of supercooling were required to activate crystallization and nucleation. Higher cooling rates resulted in melt crystallization occurring at lower temperatures with higher crystallization rates.

The crystallinity of PA11(a) decreased from 35.30% to 24.18%, and that of PA11(b) decreased from 42.81% to 35.09% when the cooling rate was increased from 05 °C/min to 15 °C. At the same time, the tensile strength of PA11(a), the tensile modulus, the flexural strength and the flexural modulus decreased by 15.24%, 14.29%, 18.09%, and 6.49%, respectively, while the elongation at break increased by 46.67%. The tensile strength of PA11(b), the tensile modulus, the flexural strength and the flexural modulus decreased by 5.70%, 5.36%, 11.65%, and 3.39%, respectively, while the elongation at break increased by 8.25%. Finally, the nadir permeability coefficients of the two materials (3.61 × 10^−13^ cm^3^·cm/cm^2^·s·Pa and 1.74 × 10^−13^ cm^3^·cm/cm^2^·s·Pa) were observed at 05 °C/min. The barrier properties were significantly improved at a lower cooling rate. The simple, rapid and economical modification of the liner material in this work provides a foothold for the manufacture of type IV hydrogen storage vessel liners.

## Figures and Tables

**Figure 1 molecules-28-02425-f001:**
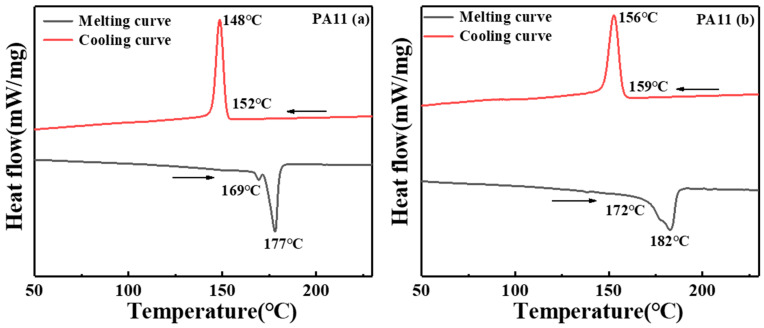
DSC curves of two PA11 materials.

**Figure 2 molecules-28-02425-f002:**
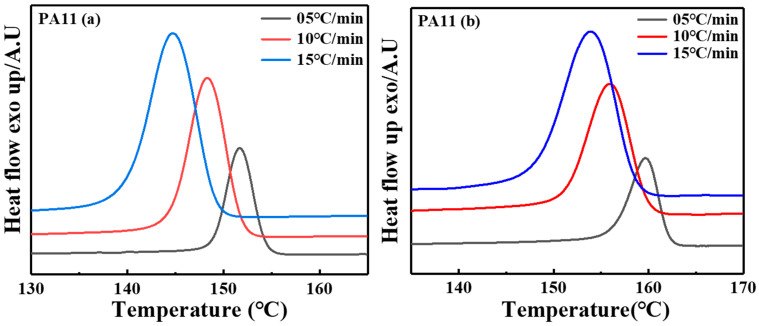
The DSC cooling curves of non-isothermal crystallization for two PA11 specimens at different cooling rates.

**Figure 3 molecules-28-02425-f003:**
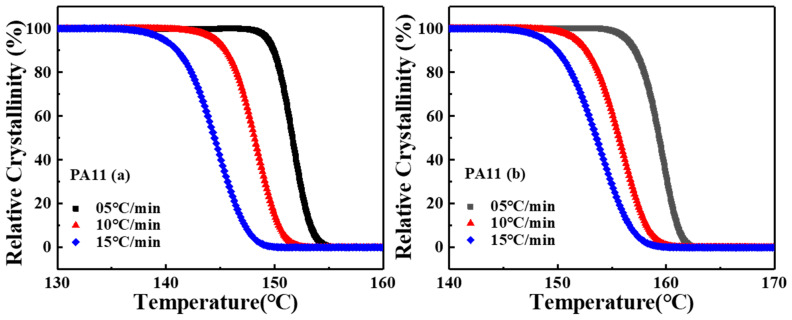
The relative degree of crystallinity versus temperature for non-isothermal crystallization of two PA11 specimens at various cooling rates.

**Figure 4 molecules-28-02425-f004:**
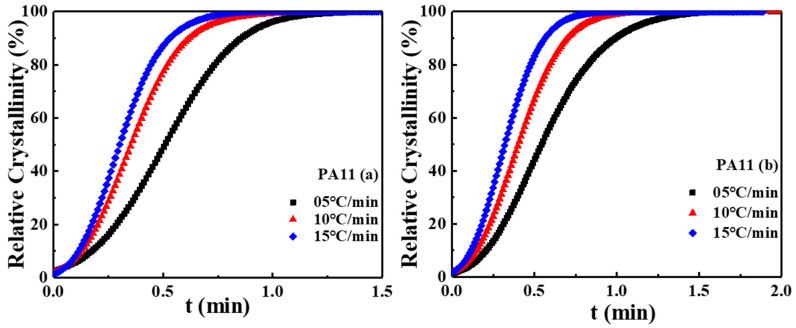
The relative crystallinity versus time for non-isothermal crystallization of two PA11 specimens at different cooling rates.

**Figure 5 molecules-28-02425-f005:**
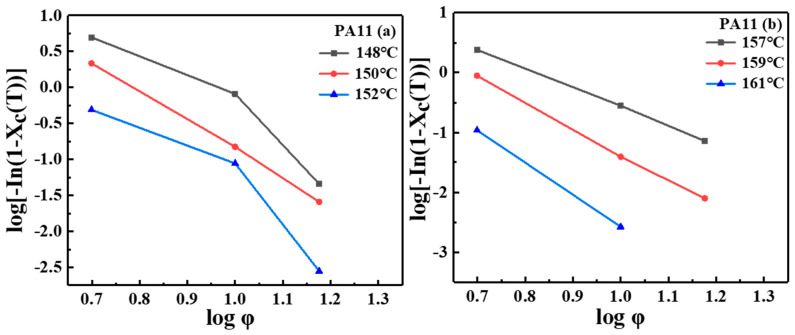
Plot of $log[-In(1-X_{c}\left(t\right))]$ versus logφ for two PA11 specimens.

**Figure 6 molecules-28-02425-f006:**
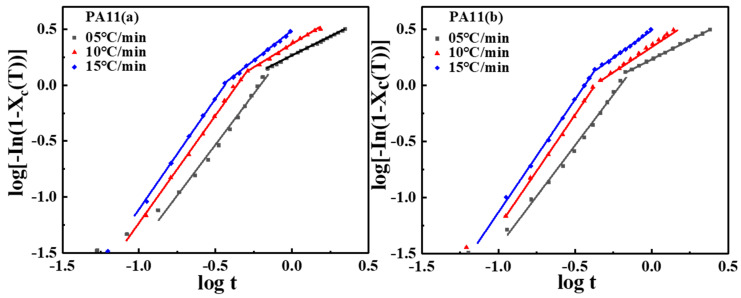
Plot of $log[-In(1-X_{c}\left(t\right))]$ versus logt for the non-isothermal crystallization process of PA11 samples.

**Figure 7 molecules-28-02425-f007:**
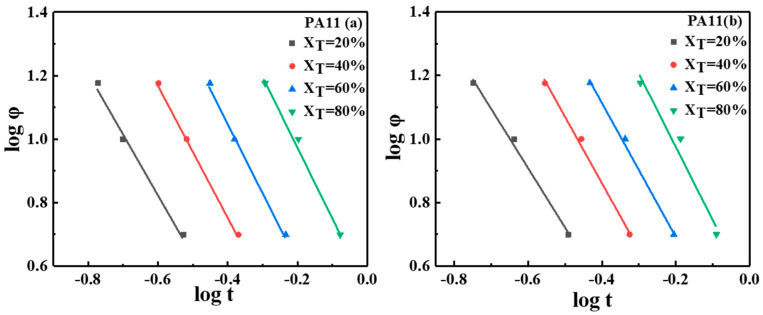
Plot of logφ versus logt at different relative crystallinity percentages.

**Figure 8 molecules-28-02425-f008:**
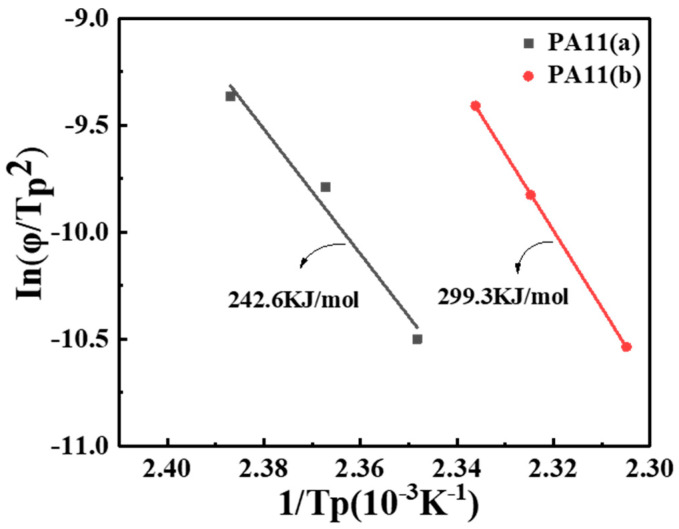
Plot of $In(\varphi /\left(T_{p}^{2}\right)$ versus $1/T_{p}$ from the Kissinger method for PA11 specimens.

**Figure 9 molecules-28-02425-f009:**
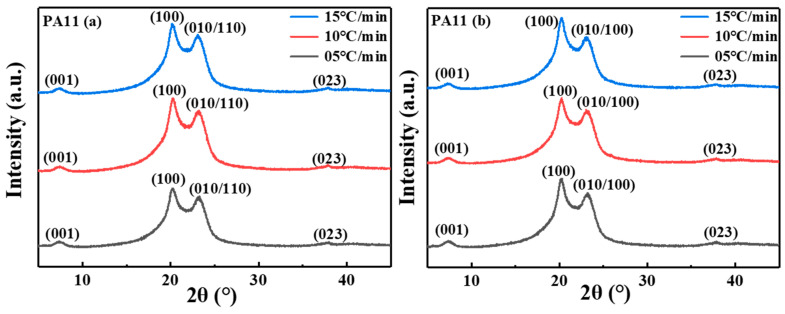
Diffractograms recorded for two PA11 crystallized samples.

**Figure 10 molecules-28-02425-f010:**
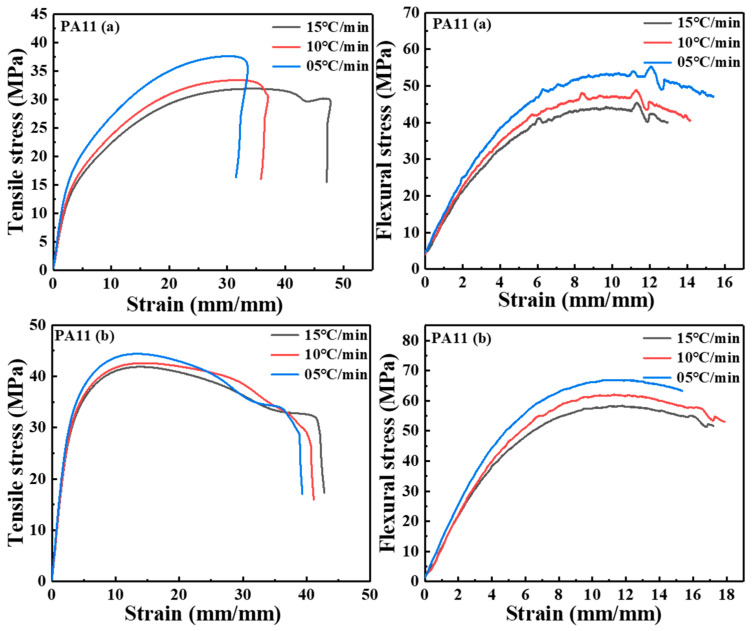
Stress–strain curves of the PA11 specimens.

**Figure 11 molecules-28-02425-f011:**
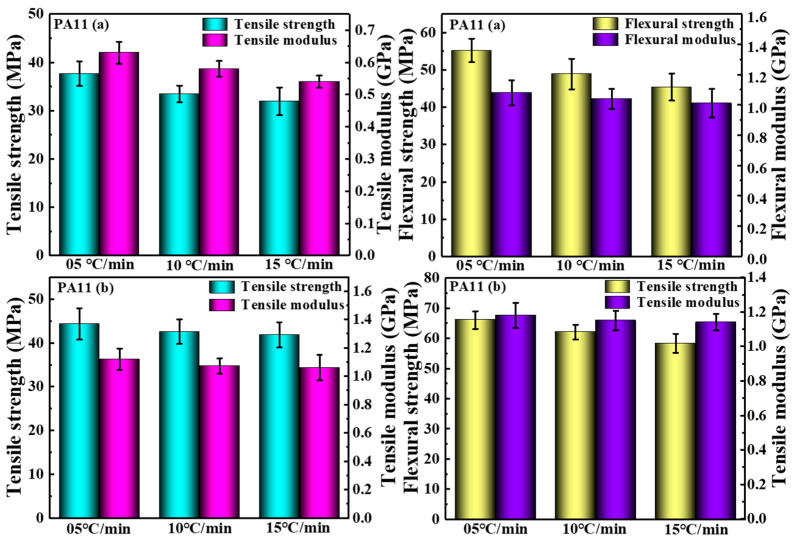
Mechanical properties test results.

**Figure 12 molecules-28-02425-f012:**
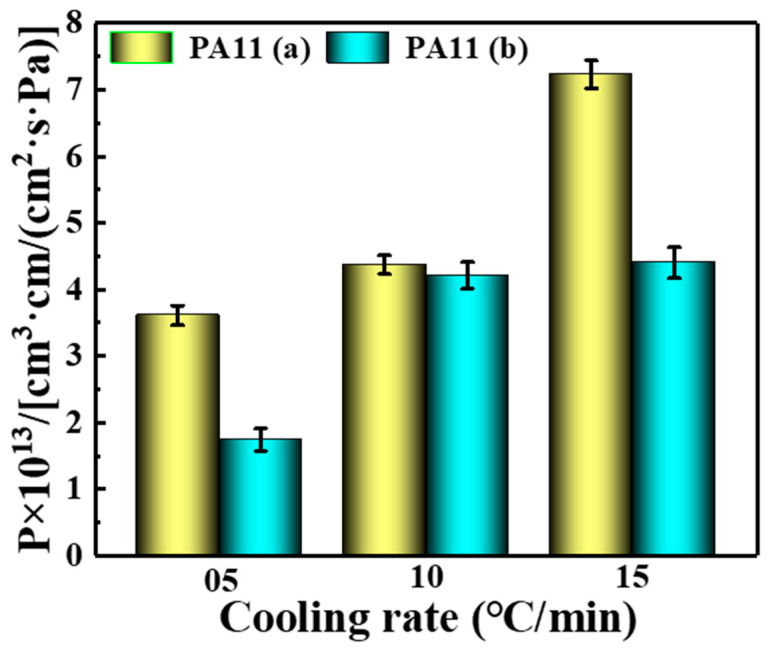
Gas permeability coefficient at different cooling rates.

**Figure 13 molecules-28-02425-f013:**
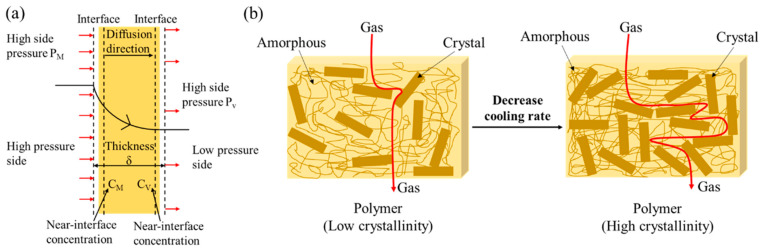
(**a**) Schematic diagram of the gas permeation process from the inner wall surface. (**b**) Gas permeation model.

**Figure 14 molecules-28-02425-f014:**
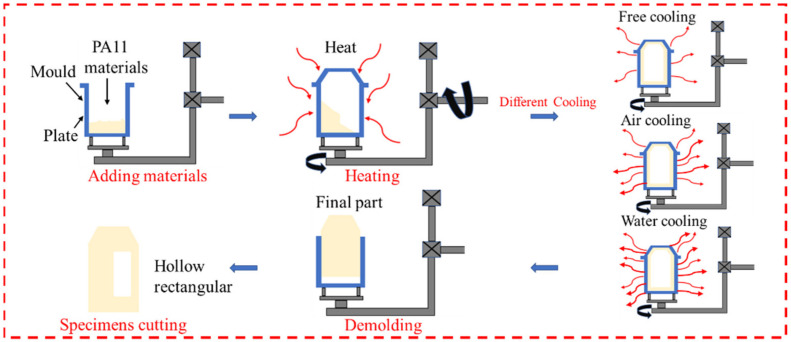
The rotational molding process of different cooling rates.

**Figure 15 molecules-28-02425-f015:**
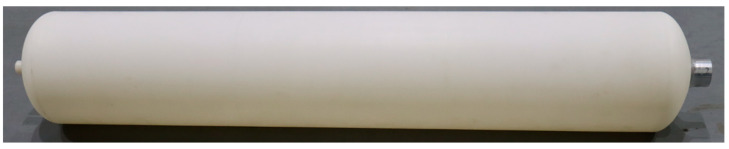
The polymer liner rotational molding specimen.

**Figure 16 molecules-28-02425-f016:**
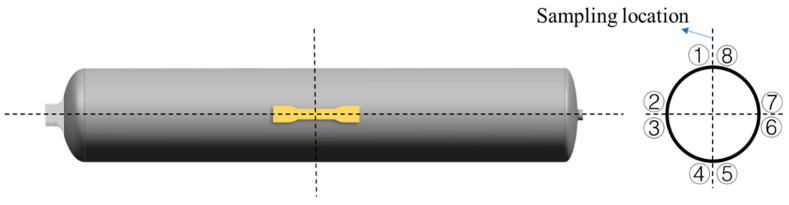
Preparation process of test strips for mechanical properties of PA11 liner materials.

**Table 1 molecules-28-02425-t001:** Non-isothermal crystallization values of two PA11 at varying cooling rates.

Sample	φ	$T_{o}$	$T_{P}$	$T_{e}$	$t_{c}$
PA11 (a)	05	154.2	151.7	149.1	1.02
	10	151.7	148.3	144.8	0.69
	15	149.0	144.8	140.3	0.58
PA11 (b)	05	162.1	159.7	156.4	1.14
	10	159.7	156.0	151.7	0.8
	15	158.4	153.9	148.5	0.66

**Table 2 molecules-28-02425-t002:** Avrami constants for all samples at various cooling rates.

φ	PA11 (a)				PA11 (b)			
	Primary Crystallization		Secondary Crystallization		Primary Crystallization		Secondary Crystallization	
	n_1_	Z_t1_	n_2_	Z_t2_	n_1_	Z_t1_	n_2_	Z_t2_
05	1.81	0.37	0.68	0.26	1.80	0.37	0.70	0.28
10	1.92	0.69	0.83	0.36	2.02	0.76	0.97	0.41
15	1.94	0.87	1.06	0.48	2.01	0.90	0.99	0.51

**Table 3 molecules-28-02425-t003:** The values of b and F(T) at different relative crystallinities.

$$X_{C\left(T\right)}$$	PA11(a)		PA11(b)	
	b	F(T)	b	F(T)
20%	1.91	0.47	1.86	0.61
40%	2.07	0.85	2.09	1.06
60%	2.17	1.53	2.11	1.87
80%	2.22	3.34	2.30	3.29

**Table 4 molecules-28-02425-t004:** $L_{hkl}$ to different crystal planes of PA11 samples with different cooling rates.

φ	PA11 (a)				PA11 (b)			
	$L_{001}$	$L_{100}$	$L_{010,100}$	$L_{023}$	$L_{001}$	$L_{100}$	$L_{010,\;\;100}$	$L_{023}$
05	2.34	3.63	3.58	2.45	2.83	3.94	3.51	2.61
10	2.22	3.32	3.04	2.31	2.44	3.87	3.69	2.53
15	2.09	3.18	2.85	2.20	2.28	3.91	3.52	2.39

**Table 5 molecules-28-02425-t005:** Rotational molding parameters.

Parameters	Values
Heating time	1800 s
Oven temperature	320 °C
Inner surface temperature	230 °C
Major axes	3RPM
Minor axes	1RPM
Cooling method	Free cooling ^1^, air cooling ^2^, water cooling ^3^
Cooling time	1500 s ^1^, 800 s ^2^, 500 s ^3^

^1^ A cooling time of 1500 s using free cooling and the cooling rate was approximately 05 °C/min. ^2^ A cooling time of 800 s using air cooling and the cooling rate was approximately 10 °C/min. ^3^ A cooling time of 500 s using water cooling and the cooling rate was approximately 15 °C/min.

## Data Availability

Not applicable.

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
