# Peer review of "The Effect of Cooling Rates on Thermal, Crystallization, Mechanical and Barrier Properties of Rotational Molding Polyamide 11 as the Liner Material for High-Capacity High-Pressure Vessels"

_molecules, 2023, doi:10.3390/molecules28062425_

Round 1

Reviewer 1 Report

The authors report the study on optimization and performance control of rotational molding polyamide 11 (PA11). They used two PA11 liner materials and changed the cooling rates of the rotational molding process, and characterized the samples with thermal, diffraction, mechanical and barrier methods.

Although the subject of the study is intensely investigated and is present in the literature, the optimization and control over the fabrication of polyamide material and its performance is still an open topic. Unfortunately, the presented study by the authors only hints at possible optimization routes without providing a systematic experimental study. 

Without a sufficient and statistically relevant number of samples and exploring a wider range of conditions, the presented study is missing control over the optimization process and enough relevant results to be publishable in this journal. The obtained results on different crystallization properties are solid and expected. Other results follow the trend, but what is missing is the amplitude of the cooling rate effect, and that is something that authors can add to the existing study to create a more rounded and systematic experimental work. In addition, the gas permeation model discussion lacks references and more analytical results to support the claims from the discussion. 

Author Response

Dear Reviewers,

Thank you very much for your promising and helpful letter. We highly appreciate your insightful comments for our manuscript, which very useful for improving the quality of the manuscript. We have modified the manuscript based on the reviewer’s comments. The revised contents are marked using the “Track Changes” function in the manuscript. Here are our responses to the comments of the reviewers:

To reviewer comments:

Comment: Although the subject of the study is intensely investigated and is present in the literature, the optimization and control over the fabrication of polyamide material and its performance is still an open topic. Unfortunately, the presented study by the authors only hints at possible optimization routes without providing a systematic experimental study. Without a sufficient and statistically relevant number of samples and exploring a wider range of conditions, the presented study is missing control over the optimization process and enough relevant results to be publishable in this journal.

Response: Thank you very much for your careful review and constructive suggestions for our manuscript. We also agree that without a sufficient and statistically relevant number of samples and exploring a wider range of conditions, the presented study is missing control over the optimization process. The cooling rate in rotational molding is less variable and really cannot be an optimized parameter. The core content of the paper is to study the effect of cooling rate on thermal properties, crystallization properties, mechanical properties and barrier properties of rotational molding polyamide 11 instead of exploring the optimization process. Based on this, we revised the title and relevant content has been revised in the manuscript, such as lines 1-10 of the abstract, 3th paragraph of introduction.

Comment: The obtained results on different crystallization properties are solid and expected. Other results follow the trend, but what is missing is the amplitude of the cooling rate effect, and that is something that authors can add to the existing study to create a more rounded and systematic experimental work.

Response: Thank you very much for your helpful suggestions. We strongly agree that a wider range of cooling rates needs to be considered. In fact, we have considered this problem when conducting the DSC test, so we did the DSC test with the cooling rate of 05 °C/min, 07 °C/min, 10 °C/min, 12 °C/min, 15 °C/min, 17 °C/min and 20 °C/min, the result trend is consistent with the manuscript trend. But, the samples for testing crystallization properties were prepared by hot-stage, and the test samples for mechanical properties were prepared by rotational experiments. None of these molding processes can achieve too wide cooling rates. Especially in rotational molding experiments, there are only three cooling methods: natural cooling, air cooling and water cooling. Investigating other cooling rates was less instructive for the final process. So, all tests used the cooling rate that can be achieved by rotational molding. Similar situations have been reported in other papers, zhang[1] studied the effect of cooling rates on crystallization and low-velocity impact behavior of carbon fiber reinforced poly (aryl ether ketone) composites. The cooling rate of DSC experiment were only studied at 1 ℃/min, 3 ℃/min,10 ℃/min and 30 ℃/min, because the cooling rate of the final compression molding test was relatively limited. Lee[2] conducted DSC and XRD tests, the cooling rate was set to 1, 10 and 20 ℃/min, because final GFPP composites were cooled by the water cooling system in hot-press machine with the different cooling rates as 1, 10 and 20 °C/min. Sota Oshima[3] conducted the DSC test with the cooling rate set to be 1, 5 and 10 °C/min, because CF/PPS composites were cooled by an air cooling system at three different cooling rates of 1, 5 and 10 °C/min.

[1] Zhang J, Liu G, An P, et al. The effect of cooling rates on crystallization and low-velocity impact behaviour of carbon fibre reinforced poly (aryl ether ketone) composites[J]. Composites Part B: Engineering, 2023, 254: 110569.

[2] Lee I G, Kim D H, Jung K H, et al. Effect of the cooling rate on the mechanical properties of glass fiber reinforced thermoplastic composites[J]. Composite Structures, 2017, 177: 28-37.

[3] Oshima S, Higuchi R, Kato M, et al. Cooling rate-dependent mechanical properties of polyphenylene sulfide (PPS) and carbon fiber reinforced PPS (CF/PPS) [J]. Composites Part A: Applied Science and Manufacturing, 2023, 164: 107250.

Comment: The gas permeation model discussion lacks references and more analytical results to support the claims from the discussion.

Response: Thank you very much for your careful review and constructive suggestions for our manuscript. We have introduced some recent references to discuss gas permeation models, such as references [47], [48], [49] [50], [51] [52], [53]and [14]. And, relevant content has been added in the revised manuscript, such as 4th paragraph of ‘Barrier properties’. We also strongly agree that more analytical results to support the claims from the discussion. For example, gas permeation needs to consider the high temperature and pressure environment. However, due to limited equipment and safety considerations, we only conduct hydrogen permeation experiments at normal temperature and pressure.

Reviewer 2 Report

Section 2: twice ‘produced by XXX  please replace XXX

I have a question on Figure 12. I have printed these out and made an overlay. This is an old fashioned but often effective way to look for differences. Nowadays things are done exclusively with computer tools. With the eye one does not see much differences, so the question is whether there are true differences and trends within the limits of accuracy (noise etcetera).  

Author Response

Dear Reviewers,

Thank you very much for your promising and helpful letter. We highly appreciate your insightful comments for our manuscript, which very useful for improving the quality of the manuscript. We have modified the manuscript based on the reviewer’s comments. The revised contents are marked using the “Track Changes” function in the manuscript. Here are our responses to the comments of the reviewers:

To reviewer comments:

Comment: Section 2: twice ‘produced by XXX’ please replace XXX

Response: Thank you very much for your careful review and constructive suggestions for our manuscript. We have replaced ‘produced by XXX’ with ‘XXX’ and relevant content has been revised in the manuscript.

Comment: I have a question on Figure 12. I have printed these out and made an overlay. This is an old fashioned but often effective way to look for differences. Nowadays things are done exclusively with computer tools. With the eye one does not see much differences, so the question is whether there are true differences and trends within the limits of accuracy (noise etcetera). 

Response: Thank you very much for your helpful suggestions. We processed the data by normalization method and compared the cooling rate data of 5°C/min and 15°C/min. We can see that the 05 °C/min curve surrounds the 15 ℃/min curve and there are differences in some crystallization peaks.

Round 2

Reviewer 1 Report

The authors addressed the raised concerns and significantly improved the manuscript. I would recommend it for publishing in its new revised form.